# Phase Angle in Head and Neck Cancer: A Sex-Differential Analysis from Biological and Clinical Behavior to Health-Related Quality of Life

**DOI:** 10.3390/biomedicines11061696

**Published:** 2023-06-12

**Authors:** Brenda-Eugenia Martínez-Herrera, Leonardo-Xicotencatl Gutiérrez-Rodríguez, Benjamín Trujillo-Hernández, Michelle-Guadalupe Muñoz-García, Luz-María Cervantes-González, Laura-Liliana José Ochoa, Javier-Andrés González-Rodríguez, Alejandro Solórzano-Meléndez, Eduardo Gómez-Sánchez, Gabriela-Guadalupe Carrillo-Nuñez, Mario Salazar-Páramo, Arnulfo-Hernán Nava-Zavala, Martha-Cecilia Velázquez-Flores, Carlos-M. Nuño-Guzmán, Mario-Alberto Mireles-Ramírez, Luz-Ma.-Adriana Balderas-Peña, Daniel Sat-Muñoz

**Affiliations:** 1Departamento de Nutrición y Dietética, Hospital General de Zona #1, Órgano de Operación Administrativa Desconcentrada (OOAD), Instituto Mexicano del Seguro Social, Aguascalientes 20270, Mexico; bren.mtzh16@gmail.com; 2Carrera de Médico Cirujano y Partero, Coordinación de Servicio Social, Centro Universitario del Sur, Universidad de Guadalajara (UdG), Ciudad Guzmán 49000, Mexico; xicotencatl.gutierrez@alumno.udg.mx; 3Doctorado en Ciencias Médicas, Facultad de Medicina, Universidad de Colima, Colima 28040, Mexico; trujillobenjamin@hotmail.com; 4Unidad de Investigación Biomédica 02 (UIBM 02), Unidad Médica de Alta Especialidad (UMAE), Hospital de Especialidades (HE), Centro Médico Nacional de Occidente (CMNO), Instituto Mexicano del Seguro Social (IMSS), Torreon 27170, Mexico; michelle.munoz8626@alumnos.udg.mx (M.-G.M.-G.); luz.cervantes7983@alumnos.udg.mx (L.-M.C.-G.); 5Carrera de Médico Cirujano y Partero, Coordinación de Servicio Social, Centro Universitario de Ciencias de la Salud (CUCS), Universidad de Guadalajara (UdG), Guadalajara 44340, Mexico; laura.jose@alumnos.udg.mx (L.-L.J.O.); javier.gonzalez5722@alumnos.udg.mx (J.-A.G.-R.); 6Comisión Interinstitucional de Formación de Recursos Humanos en Salud, Programa Nacional de Servicio Social en Investigación 2021, Demarcación Territorial Miguel Hidalgo 11410, Mexico; 7Comité de Cabeza y Cuello, UMAE, Hospital de Especialidades, Centro Médico Nacional de Occidente, Instituto Mexicano del Seguro Social, 1000 Belisario Domínguez, Guadalajara 44340, Jalisco, Mexico; dr_solorzano@yahoo.com.mx; 8Departamento Clínico de Oncología Radioterapia, Servicio Nacional de Radioneurocirugía, División de Oncología Hematología, UMAE, Hospital de Especialidades, Centro Médico Nacional de Occidente, Instituto Mexicano del Seguro Social, Guadalajara 44340, Jalisco, Mexico; 9Cuerpo Académico UDG CA-874 “Ciencias Morfológicas en el Diagnóstico y Tratamiento de la Enfermedad”, Guadalajara 44340, Jalisco, Mexico; eduardo.gsanchez@academicos.udg.mx; 10División de Disciplinas Clínicas, Centro Universitario de Ciencias de la Salud (CUCS), Universidad de Guadalajara (UdG), 950 Sierra Mojada, Building N, 1st Level, Guadalajara 44340, Jalisco, Mexico; carlosnunoguzman@hotmail.com; 11Departamento de Microbiología y Patología, Cuerpo Académico UDG CA-365 “Educación y Salud” Centro Universitario de Ciencias de la Salud (CUCS), Universidad de Guadalajara (UdG), Guadalajara 44340, Jalisco, Mexico; gabriela.cnunez@academicos.udg.mx; 12Academia de Inmunología, Departamento de Fisiología, Centro Universitario de Ciencias de la Salud (CUCS), Universidad de Guadalajara (UdG), Guadalajara 44340, Jalisco, Mexico; mario.sparamo@academicos.udg.mx; 13Unidad de Investigación Epidemiológica y en Servicios de Salud, Centro Médico Nacional de Occidente, Órgano de Operación Administrativa Desconcentrada (OOAD) Jalisco, Instituto Mexicano del Seguro Social, Guadalajara 44340, Jalisco, Mexico; navazava@yahoo.com.mx; 14Programa Internacional Facultad de Medicina, Universidad Autónoma de Guadalajara, Zapopan 45129, Jalisco, Mexico; 15Servicio de Inmunología y Reumatología, División de Medicina Interna, Hospital General de Occidente, Secretaria de Salud Jalisco, Zapopan 45170, Jalisco, Mexico; 16Departamento Clínico de Anestesiología, División de Cirugía, UMAE, Hospital de Especialidades, Centro Médico Nacional de Occidente, Instituto Mexicano del Seguro Social, Guadalajara 44340, Jalisco, Mexico; c.velazquez@academicos.udg.mx; 17Departamento de Morfología, Centro Universitario de Ciencias de la Salud (CUCS), Universidad de Guadalajara (UdG), Guadalajara 44340, Jalisco, Mexico; 18Departamento Clínico de Cirugía General, División de Cirugía, UMAE, Hospital de Especialidades, Centro Médico Nacional de Occidente, Instituto Mexicano del Seguro Social, Guadalajara 44340, Jalisco, Mexico; 19Servicio de Cirugía General, OPD Hospital Civil de Guadalajara “Fray Antonio Alcalde”, Guadalajara 44280, Jalisco, Mexico; 20División de Investigación en Salud, UMAE, Hospital de Especialidades, Centro Médico Nacional de Occidente, Instituto Mexicano del Seguro Social, Guadalajara 44340, Jalisco, Mexico; mario.mirelesr@imss.gob.mx; 21Departamento Clínico de Oncología Quirúrgica, División de Oncología Hematología, UMAE, Hospital de Especialidades, Centro Médico Nacional de Occidente, Instituto Mexicano del Seguro Social, Guadalajara 44340, Jalisco, Mexico

**Keywords:** head and neck cancer, phase angle, sex, C-reactive protein/albumin ratio, malnourishment, bioimpedance analysis, quality of life

## Abstract

Head and neck cancer (H&NC) is a diverse category of tumors related to malignancies in the common aerodigestive pathway, with high metabolic rate, poor nutritional and treatment outcomes, and elevated mortality despite the best standard treatment. Herein, we focus on determining how the phase angle (PA) differs across sex as a predictor of poor prognosis, low quality-of-life (QoL) scores, and mortality in patients with head and neck cancer. This follow-up study presents a sex-differential analysis in a prospective cohort of 139 head and neck cancer patients categorized by sex as male (n = 107) and female (n = 32). Patients were compared in terms of nutritional, biochemical, and quality-of-life indicators between low and normal PA in women (<3.9° (n = 14, 43.75%) and ≥3.9°) and men (<4.5° (n = 62, 57.9%) and ≥4.5°). Our results show that most patients were in locally advanced clinical stages (women: n = 21 (65.7%); men: n = 67 (62.6%)) and that patients with low PA had a lower punctuation in parameters such as handgrip strength, four-meter walking speed, albumin, C-reactive protein (CRP), and CRP/albumin ratio (CAR), as well as the worst QoL scores in functional and symptomatic scales in both the male and female groups. A comparison between sexes revealed significant disparities; malnourishment and tumor cachexia related to an inflammatory state was more evident in the women’s group.

## 1. Introduction

Head and neck cancer (H&NC) is a highly prevalent malignant disease in Asian countries, with around 900,000 total cases, 550,000 newly diagnosed cases per year, and 400,000 deaths annually [1]. Males are predominantly affected, with the male/female ratio ranging from 2:1 to 4:1 [2]. The geographic distribution of N&NC is determined by habits considered risk factors, such as tobacco, alcohol, and betel nut consumption. Other identified risk factors are opium use; viral infections (Epstein–Barr virus, human papillomavirus, herpes simplex virus, hepatitis C virus, and even human immunodeficiency virus); radiation; occupational exposure to perchloroethylene, pesticides, asbestos, and polycyclic aromatic hydrocarbons; dietary aspects, such as high levels of nitrites; mouthwashes with alcohol derivates; and poor oral hygiene and periodontal disease [3,4,5,6,7].

Head and neck cancer describes a group of malignancies, mainly squamous histologic varieties (over 90%), involving various anatomic sites and subsites, including the oral cavity, pharynx, larynx, nasal cavity, and paranasal sinuses or salivary glands. This condition affects the initial symptoms, progression, and malnourishment rate (Overview of the diagnosis and staging of head and neck cancer in up to date).

All these tumors share biological characteristics [8], with a high mortality rate in both sexes compared with other malignant neoplasms [9,10,11,12], even in the patients treated in our facility care center, where the rate reached 70% [13].

A 2020 report on the global burden of cancer by The Global Cancer Observatory (GLOBOCAN) [6] differentiated the disease prevalence according to anatomical site and subsite, with lip and oral cavity cancer accounting for 377,713 new cases in both sexes of all ages. The incidence in males accounted for 264,211 new cases versus 113,502 new cases in females. The number of male deaths was 125,022 (cumulated risk: 0.32), whereas the number of female deaths was 52,735 (cumulated risk: 0.12) [14]. The differential biological behavior between males and females is crucial to understanding the role of sex-specific biological characteristics in disease development associated with structural, hormonal, and immunological influences [15].

Sex differences in cancer vary considerably through mechanisms that impact incidence, prognosis, treatment, and survival [16], an outcome of differences in metabolism, sex-hormone-dependent systems, cell cycle control, and immune response specific to biological sex, which may be the reason for these repercussions [16,17].

In 2020, De Courcy et al. postulated about the radiotherapy response and side effects related to sex-differentiated biological behavior, which was radiogenomically investigated [18].

The radiotherapy basement refers to the use of ionizing radiation to ionize and oxidate biological molecules inside cells, resulting in the generation of free radicals and DNA damage due to radiation itself or through free oxygens radicals, causing membrane damage, senescence with alterations in cellular reproduction, apoptosis, or mitotic catastrophe [19].

All these mechanisms involve approximately 52% of malignant neoplastic cells causing tumor lysis. Another benefit is enhancement of innate immune system function associated with tumor regression during metastasis phenomena [20].

The adverse events associated with radiotherapy are acute and chronic with respect to the anatomic site and subsite. In H&NC, those events affect the vicinity of the aerodigestive tract, including superficial skin damage, mucositis in the oral cavity and esophagus, and, in severe cases, stenosis or perforation and fistulae, as well as osteoradionecrosis depending on the radiation dose, the volume of irradiated healthy tissue and irradiated tissue, and even the sex of the patient. According to Alsbeth et al. and Barnett et al., sex-related variations occur in the ability to recognize and repair DNA double-strand breaks as part of a differential treatment response [21,22,23].

In 2019, Benchetrit L et al. described the behavior of a population with squamous H&NC between 1985 and 2015. Women represented 26% of the total subjects, and a lower percentage of women were candidates to receive chemotherapy or chemoradiotherapy, considering their clinical condition. The clinical stages of H&NC were similar in proportion to the studied men [24].

H&NC is also one of the most aggressive types of cancer, causing patients to undergo life-changing interventions that impact their individual and social well-being and reduce function and quality of life (QoL) [25,26]. H&NC frequently leads to severe nutritional problems that result from the disease itself, even before treatment or treatment sequelae of surgery with extensive resection and reconstruction, including the aerodigestive tract, radiation, or acute or chronic effects of chemotherapy [27].

The clinical course of the disease also comprises many complications, such as dysphagia, swallowing difficulties, pain, inadequate dietary intake, hemoptysis, dyspnea, and oral mucositis, which impact the ability of patients to speak, breathe, and eat, with a high association with malnutrition [28,29].

Aerodigestive symptomatology interferes with food intake, which, in addition to systemic inflammation, leads to anorexia, loss of skeletal muscle mass and functionality, and adipose tissue depletion [16,17]. Patients with H&NC also exhibit poor overall survival rates [30].

The natural history of H&NC, as well as its treatments, impairs nutritional status; thus, anorexia and cachexia resulting from the therapeutic approach are associated with increased morbidity and mortality rates, QoL deterioration, and elevated medical treatment costs.

The high metabolic rate of H&NC, combined with the abovementioned factors, contributes to malnutrition-related weight loss [31], loss of function, and a high rate of adverse outcomes, even with the best available treatment, leading to increased morbidity and mortality rates [32,33].

The criteria related to malnutrition include biochemical and anthropometric markers [34]. Body composition can be used to identify a patient’s nutritional condition by dividing body weight into several compartments to obtain information about the patient’s health and functionality [34,35].

Nutritional phenotype diagnosis by bioelectrical impedance analysis (BIA) is a body composition assessment tool used in clinical settings to detect weight loss and muscle mass changes and determine the integrity of cell membranes using phase-angle (PA) analyses [36,37,38,39].

BIA is a simple, non-invasive technique that estimates body composition by measuring the opposition (impedance) to a current passing through the body. Impedance is the result of two components: resistance (R), which is the opposition to the flow of an alternating current through intra- and extracellular ionic solutions, and reactance (Xc), which is the delay in conduction because of capacitance by cell membranes and tissue interfaces. Capacitance conditions a phase shift or PA that is quantified as an arc tangent ((Xc/R) * 180°/π) [40]. PA is a biological marker of cellular health, reflecting membrane integrity, cell mass, and hydration status. PA has been proven to be a suitable predictor of morbidity and mortality in various patient groups [41,42].

PA data relate to functionality, QoL, and even the risk of complications and mortality [10]; their use is based on estimation of the integrity of cell membranes, serving as a critical parameter for prediction of clinical and functional outcomes [43]. Moreover, these measures predict patient functionality and mortality [44]. Therefore, it is possible to identify the role of skeletal muscle mass loss and lean mass loss in patients with H&NC, predisposing this population to sarcopenia and dynapenia.

PA ranges from 4 to 10 in healthy individuals [39]. Low PA values are related to damage to cell membrane integrity, leading to cell death. In contrast, moderate or high values are related to cell membrane integrity and an appropriate water ratio in the intracellular-to-extracellular spaces, which can predict body mass [32,45].

Therefore, oncologic care for H&NC patients should focus on the importance of prognosis and prediction in decision making and treatment approaches that contribute to the best response and minimal toxicity of treatment [46].

The high mortality and the poor nutritional response of patients undergoing treatment for this type of cancer are the factors determining the sex-specific variation in PA as a predictor of poor prognosis and mortality in patients with head and neck cancer.

In this report, we emphasize the sex-differential behavior of nutritional status, functionality, and mortality in H&NC patients.

## 2. Materials and Methods

### 2.1. Patients

The current study presents a sex-differential analysis in a prospective cohort of 139 patients with H&NC categorized by sex as male (n = 107) and female (n = 32), all of whom had a biopsy to confirm their diagnosis. They were recruited to participate in the study at an Instituto Mexicano del Seguro Social (IMSS) tertiary care hospital in Guadalajara, Mexico. All patients provided consent in writing, and the study was approved by our institutional review board (Comisión Nacional de Investigación en Salud del IMSS). The processes were carried out in accordance with the requirements of the Helsinki Declaration.

Patients with more than one malignant neoplasm, autoimmune diseases, chronic illnesses in the lung or kidney, or any contraindication to perform bioelectrical impedance analysis were excluded. We retrieved the patients’ clinical features, clinical stage, anatomical localization of the tumor, treatment, and lab results from medical records.

The following requirements had to be followed to complete the anthropometric analysis: the patients had to fast for eight hours and be free of any objects and conditions that would interfere with the BIA (metal prosthesis, shoes, socks, fixtures, electronic implants, severe edema, limb amputations, weight greater than 300 kg, etc.).

### 2.2. Anthropometric Analysis

An SECA 213 instrument (Seca, Hamburg, Germany) was used to measure weight (kg) and height (m), whereas a SECA 514 bioelectrical impedance device (Seca, Germany) was used to assess body composition and PA.

The body mass index (BMI) and skeletal muscle mass index (SMMI) were calculated by dividing the body weight or total skeletal muscle mass (kg) by the height squared (m^2^). The patients were then classified into three categories based on nutritional phenotype:
Non-sarcopenic group (NSG):
(a)Women with SMMI ≥ 6.42 kg/m^2^ and BMI < 25 kg/m^2^;(b)Men with SMMI ≥ 8.86 kg/m^2^ and BMI < 25 kg/m^2^.
Sarcopenic group (SG):
(a)Women with SMMI < 6.42 kg/m^2^ and BMI < 25 kg/m^2^;(b)Men with SMMI < 8.87 kg/m^2^ and BMI < 25 kg/m^2^.Sarcopenic obesity group (SOG):
(a)Women with SMMI < 6.42 kg/m^2^ and BMI ≥ 25 kg/m^2^;(b)Men with SMMI < 8.87 kg/m^2^ and BMI ≥ 25 kg/m^2^.


A dietitian validated the body composition analysis and the nutritional phenotypic classification. An mBCA SECA 514 bioelectric impedance device (Seca, Germany) was used to determine patient weight, phase angle, total skeletal muscle mass, and whole-body fat percentage, with data used to calculate the body mass index (BMI), skeletal muscle mass index (SMMI), and fat mass index (FMI). All of these indicators were used to calculate patient outcomes.

BMI was calculated as described by the World Health Organization. We obtained the SMMI by dividing the total skeletal muscle mass (kg) by the height squared (m^2^). The SMM-to-patient weight ratio was used to calculate the proportion of total weight corresponding to muscle mass, which was normalized by dividing by the square of height. Handgrip strength was assessed using a Jamar Plus+ Digital hand dynamometer (Patterson Medical Supply, Cedarburg, WI, USA). According to the American Association of Hand Therapists, patients held the device and compressed it with maximum force to obtain a maximum contraction. The test was repeated three times for each hand, with one-minute rest intervals between measurements. The highest result of all tests was recorded [47].

The EORTC QLQ-C30 v.3 (validated for Mexican Spanish; Brussels, Belgium) questionnaire was conducted to assess HRQoL. The instrument consists of six multi-item scales (related to patient functioning) and nine single-item scales (describing the severity of cancer-related symptoms). The EORTC QLQ-H&N35 supplementary module for H&NC patients was also used, which consists of 35 questions, 7 multi-item symptom scales, and 11 single-item symptom scales described by the EORTC Scoring Manual [48].

### 2.3. Statistical Analysis

Statistical analysis was performed with the IBM^®^ SPSS^®^ Statistics version 29 software package (Armonk, NY, USA).

Results are presented as means ± standard deviation (SD) for variables with normal distributions. Non-parametric variables are described as medians (interquartile intervals (IQIs)).

Categorical variables are expressed as numbers and percentages relative to the total. Pearson’s chi-squared tests were performed to assess differences between the two groups (Fisher’s tests if the estimated values were <5), and one-way ANOVA and Kruskal–Wallis tests with Bonferroni correction were used to assess differences between the three groups. Pearson’s correlation or Spearman’s Rho was calculated to determine the relationship, depending on the variable type. Survival analysis was carried out using the Kaplan–Meier method. Analyses were two-sided, and a *p* value of <0.05 was considered significant. Cronbach’s alpha value was used to determine reliability in the multi-item scales of the EORTC questionnaires.

## 3. Results

This study included 139 H&NC patients (107 (76.98%) men and 32 (23.02%) women), with a two-year follow-up. Population analysis showed a perspective between sexes across clinical, anthropometric, and biochemical parameters (Table 1). Significant differences in age (*p* = 0.012), phase angle (*p* = 0.035), handgrip strength, SMMI, hemoglobin, and total body fat were observed (all with *p* < 0.001; Table 1). We found differences around the anatomical location of the tumor; in women, a higher percentage presented a tumor in the oral cavity (n = 16; 50%), and for men, the larynx represented the most affected anatomical site (n = 51; 47.7%; *p* = 0.04).

The median age of the 32 women was 59 (44–69) years. The predominant anatomical location was the oral cavity (n = 16; 50%), with histological diagnosis of squamous or epi-dermoid carcinoma (n = 22; 68.8%) and clinical stage IV (n = 14; 43.8%).

The median age of the 107 men was 67 (60–73) years. The predominant anatomical location was the larynx (n = 51; 47.7%), with histological diagnosis of squamous or epidermoid carcinoma (n = 101; 94.4%) and clinical stage IV (n = 49; 45.8%).

### 3.1. Population Characteristics by Gender

#### 3.1.1. Comparison between Low and Normal PA in Women

We divided the women’s group into low PA (<3.9°) and normal PA (≥3.9°). In both groups, most of the tumors were of squamous histologic variety. In more than 50% of both groups, the tumors were in clinical stages III and IV. Sarcopenia was more common in women with low PA. A proportion of 71.4% of women with low PA were not treated with surgery, and 50% were managed using chemoradiotherapy (Table 2). Tumor location predominance in the oral cavity was observed in women with normal PA (n = 10; 55.6%), without any specific predominance in low-PA women.

#### 3.1.2. Comparison between Low and Normal PA in Men

In men, the cutoff for PA was 4.5, and the most affected anatomical area was the larynx, without specific distribution related to PA values. In both groups, the predominant histologic type was squamous, and the dominant clinical stage was CS IV. Sarcopenia was present in 46.8% of patients with low PA, versus 26.7% for normal-PA men (*p* = 0.023). Men in the low-PA group were not treated with surgery in 69.4% of cases, compared to 51.1% in normal-PA men (*p* = 0.043). We did not identify other differences conditioned by PA values in the male group.

### 3.2. Anthropometrical and Biochemical Indicators

#### 3.2.1. Comparison between Low and Normal PA in Women

In the group of H&NC women, we observed differences in terms of age (64 ± 16.5 versus 51.8 ± 14.8; *p* = 0.013), handgrip strength (17 ± 4.5 kg/cm^2^ versus 22.6 kg/cm^2^; *p* = 0.024, respectively), four-meter walking speed (0.7 ± 0.3 versus 0.9 ± 0.2 m/seg), phase angle (3.2 ± 0.6° versus 4.6 ± 0.5°; *p* < 0.001), and SMMI (5.5 ± 2.5 versus 7.1 ± 1.7 kg/m^2^ BS; *p* = 0.041) between low and normal PA angle (Table 3).

Concerning biochemical markers in low and normal PA H&NC women, C-reactive protein (27.1 (19–48.5) mg/dL versus 10.4 (3.6–27); *p* = 0.027) and albumin levels (3.8 ± 0.6 versus 4.3 ± 0.3 g/dL; *p* = 0.011) showed significant differences (Table 3).

#### 3.2.2. Comparison between Low and Normal PA in Men

For H&NC men, the differences observed between low and normal PA relevant in terms of age (69 ± 11 versus 60 ± 10; *p* < 0.001), handgrip strength (25.3 ± 8 versus 33 ± 7.2; *p* < 0.001), gait speed (0.75 ± 0.2 versus 0.95 ± 0.2; *p* > 0.001), phase angle (3.7 ± 0.6 versus 5.2 versus 0.5; *p* < 0.001), hemoglobin (12.9 ± 2 versus 14.3 ± 1.7 g/dL; *p* < 0.001), absolute lymphocyte count (1406 (1120–2015) versus 1739 (1311–2515); *p* = 0.024), albumin (4.1 ± 0.5 versus 4.3 ± 0.3; *p* < 0.001), and C-reactive protein (16.3 (5–27) versus 12 (3–21); *p* = 0.022; Table 3).

### 3.3. Health-Related Quality-of-Life Indicators

#### 3.3.1. Comparison between Low and Normal PA in Women

We found profound alterations in various scales for the EORTC QLQ-C30 and EORTC QLQ-CX24 questionnaires, with significant differences in the scores for global health status/QoL (58.3 versus 75; *p* = 0.049), physical functioning (50 versus 93.3; *p* < 0.001), fatigue (61.1 versus 22; *p* = 0.014), loss of appetite (50 versus 0; *p* = 0.014), swallowing (33.3 versus 4.2; *p* = 0.020), trouble with social contact (33.3 versus 0.0; *p* = 0.030), teeth (100 versus 10.3; *p* = 0.020), and sticky saliva (100 versus 0.0; *p* = 0.027) (see Table 4).

#### 3.3.2. Comparison between Low and Normal PA in Men

In the H&NC men group, the findings of the EORTC QLQ-C30 questionnaire were significant on the scales of global health status/QoL (69.8 versus 83.3; *p* = 0.006), physical functioning (73.3 versus 93.3; *p* < 0.001), role functioning (83.3 versus 100; *p* = 0.032), fatigue (44.4 versus 11.1; *p* < 0.001), loss of appetite (0.0 (0.0–33.3) versus 0.0 (0.0–0.0); *p* = 0.013). On the EORTC QLQ-CX24 questionnaire, we observed significant differences in swallowing (25 versus 8.3; *p* = 0.010), sensory problems (16.7 versus 0.0; *p* = 0.023), trouble with social eating (19.4 versus 8.3; *p* = 0.031), mouth opening (23 versus 0.0; *p* = 0.043), and pain killers (100 versus 0.0; *p* = 0.022) (see Table 4).

### 3.4. Survival Status by Sex and by PA

#### 3.4.1. Comparison between Low and Normal PA in Women

In the women’s groups considering low and normal PA, the survival rate showed statistical differences between groups. The women with a low phase angle had 71.4% two-year mortality, whereas for normal phase angle women, the mortality rate was 27.8% (*p* = 0.046). The mean survival time was shorter (14.25 (95% CI, 7.8–20.7) months) in the low-PA group than in the normal-PA group (30.9 (95% CI, 23.5–38.5) months; *p* = 0.007, Table 5; Figure 1).

#### 3.4.2. Comparison between Low and Normal PA in Men

For the male group, the mortality rate in patients with PA lower than 4.5° was 53.2%, compared with the group of male patients with a PA over 4.5°, for which the mortality rate was 26.7% (*p* = 0.022). The survival time was also shorter for patients with low PA (22.5 (18–27) versus 35.2 (29.5–40.8) months; *p* = 0.008, Table 5; Figure 1).

The survival analysis by sex also reflected significant differences (23.7 (17.6–29.8) versus 28.7 (24.7–32.6) months, *p* = 0.001).

## 4. Discussion

The etiology, treatments, and clinical outcomes of oncologic illness are profoundly influenced by several biologic determinants, such as sex, age, physical function, and body composition [49,50]. A previous literature review discussed the evidence for sex dimorphism; men and women presented sex-specific regulation concerning several illness states [49].

In this instance, we discussed the involvement of biological issues and their relationship with development, treatment, and outcomes to comprehend sex-specific variations in cancer-induced gene regulation [49,50].

H&NC patients are affected by skeletal muscle loss. This situation might be caused by sex differences in metabolism, cellular function, immunological response, inflammatory phenomena, and stresses contributing to cachexia. Several mechanisms may be involved in the generation of cellular depletion that condition metabolic, inflammatory, and anthropometrical changes, including muscular atrophy and dynapenia, as demonstrated by handgrip strength (0.79 versus 0.82; *p* < 0.001) and the presence of sarcopenia (n = 52; 72.7%) in both sexes (Table 1), as well as four-meter walking speed between men with low and normal PA (0.75 versus 0.95; *p* < 0.001) [49,51]. The comparison of anthropometric and biochemical characteristics between sexes revealed significant disparities. The existence of muscle mass reduction because of SMMI is related not only to malnourishment but could also be the consequence of tumor cachexia; this phenomenon is supported by higher levels of C-reactive protein in men and women with low phase angle.

A scoping review considered data from 76 studies identified through a systematic literature search and published over six years; the prevalence of sarcopenia ranged from 3.8% to 78.7% [52]. Additionally, sarcopenia was found to have a substantial and unfavorable influence on functional, psychosocial, QoL, and survival outcomes in H&NC [52]. In contrast to our results, the prevalence of sarcopenia affects women (n = 11; 34.4%) and men (n = 41; 38.3%) and was significantly associated with handgrip strength (*p* < 0.001), four-meter walking speed (*p* < 0.001), and phase angle (*p* < 0.001) in men and grip strength (*p* = 0.024), four-meter walking speed (*p* = 0.004), and phase angle (*p* < 0.001) in women. Finally, our data revealed that women cancer patients have a higher prevalence of cachexia, muscle wasting, and poorer outcomes than men.

BIA assists in identifying changes in body composition through PA indicators, the contribution of fluids and cell components, and membrane cell integrity in the human body [53]. A decrease in PA indicates cell death or an alteration of cell membrane integrity [53]. Relevant publications demonstrate that PA assists in evaluating the nutritional and health state of cancer patients, supplying essential information that can be used as a prognostic factor [54].

Władysiuk et al. conducted a cohort study in which 75 H&NC presurgical patients were divided into PA < 4.73° and ≥ 4.73°. They observed that the odds of shorter survival were significantly higher in patients with PA < 4.73° compared to the rest of the patients (19.6 months vs. 45 months; *p* = 0.048) [55].

A study by Yamanaka et al. [56] in Asian patients suggested a 4° PA for women and 4.6° PA for men as a predictive reference value; the low-PA group had a higher risk of poor three-year survival (*p* = 0.005). Both demonstrated biological behavior similar to the pattern observed in our studied population; however, the cutoff for the Asian study was close to the characteristics that we found in our male and female patients, in which we used reference PA values of <3.9° and ≥3.9° in women and <4.5° and ≥4.5° in men. Women with low PA had a shorter survival time (14.25; (7.8–20.7) months) in comparison to the normal-PA group (30.9 (23.5–38.5)). Similar significant differences were observed in men, with reduced survival reported in the low-PA group (22.5 (18–27)) compared to the normal-PA group (35.2 (29.5–40.8)).

In our study, we found significant differences in low- and normal-PA H&NC women in terms of C-reactive protein (27.1 (19–48.5) mg/dL versus 10.4 (3.6–27); *p* = 0.027) and albumin levels (3.8 ± 0.6 versus 4.3 ± 0.3 g/dL; *p* = 0.011), which may be associated with the low PA observed in patients, a marker related to damage to cell membrane integrity, leading to cell death and, therefore, serving as a predictive marker of the functionality and mortality in association with poor outcome and prognosis.

In a cohort study, Harada et al. reviewed the records of 543 patients diagnosed with esophageal squamous cell carcinoma who underwent subtotal esophagectomy, collecting the results of CPR in blood tests performed on postoperative days. They found that CRP levels were highest on day 3. CRP levels after day 3 correlated with major complications, as well as day 7/8 high CRP levels (>3.52), combined with postoperative survival, which was significantly associated with poor prognosis (hazard ratio: 1.67; 95% CI: 1.14–2.43; *p* = 0.008), proving that CRP has potential prognostic value for patients with this diagnosis after esophagectomy [57].

Novel reports explain that specific biochemical markers emerge as an inflammation reflex. Some consider them valuable tools to evaluate prognosis in patients diagnosed with various malignancies associated with tumor cachexia [56].

Evidence shows that inflammatory response is highly associated with poor outcomes in patients diagnosed with different types of cancer, including H&NC [58], which appears to be a suitable marker for predicting survival in patients diagnosed with oral squamous cell carcinoma [59].

The CAR in Asian patients with oral squamous cell carcinoma was demonstrated to be an excellent prognosis marker. According to our results, we estimated the CAR in men and women according to PA classification. We found significant differences between females and males between low and normal PA groups (0.23 versus 0.66; *p* = 0.020, and 0.36 versus 0.28; *p* = 0.015, Table 3) [56].

Kruse et al. postulated that the C-reactive protein might be a relevant sign of chronic inflammation, serving as a prognostic marker for patients with cancer, reinforcing the findings mentioned above [60].

Finally, evaluating QoL allowed us to determine patients’ perception of their conditions as the disease progressed and was treated, as well their functional status affectation [61]. To this end, we used the questionnaires (1) the EORTC QLQ-C30 (30 questions; 6 multi-item scales related to functioning and nine single-item scales describe the severity of symptoms), (2) the EORTC QLQ-H&N35 module for H&NC patients (35 questions; 7 multi-item symptom scales, and 11 single-item symptom scales) questionnaires (validated in Mexican Spanish). The EORTC scoring system leads us to understand the functional symptoms of differential behavior according to sex [62].

The results emphasize that low PA affects QoL and global health. In women, physical function, fatigue, and loss of appetite are coincidental with a low PA and loss of muscle mass, in addition to impacting swallowing, social contact capacity, tooth state, and sticky saliva, all of which are related to functionality in women.

In the group of males, patient perception was a profound functional affectation on global health status, physical, and role functioning, and, similarly to the female group, fatigue, loss of appetite, and swallowing were also affected. According to men’s perception of low PA and loss of muscle mass, other relevant aspects include sensory issues, trouble with social eating, mouth opening, and pain management, reaching higher scores than women.

The role of sex in the perception of QoL is evident; however, there is currently a lack of differentiated information according to sex.

## 5. Conclusions

The comparison of anthropometric and biochemical characteristics between sexes reveals significant disparities. The existence of muscle mass reduction because of SMMI is more evident in women, but in terms of HRQoL, men appear to be more profoundly affected, especially in terms of the symptom scales.

The results observed in our population are associated with the incidence of malnourishment and tumor cachexia related to an inflammatory state triggered by malignant neoplasms in H&NC patients. PA was used as a significant prognostic, HRQoL, and survival indicator. PA is also linked the CAR, a novel outcome indicator for various diseases based on the systemic inflammatory response of patients with this condition.

## Figures and Tables

**Figure 1 biomedicines-11-01696-f001:**
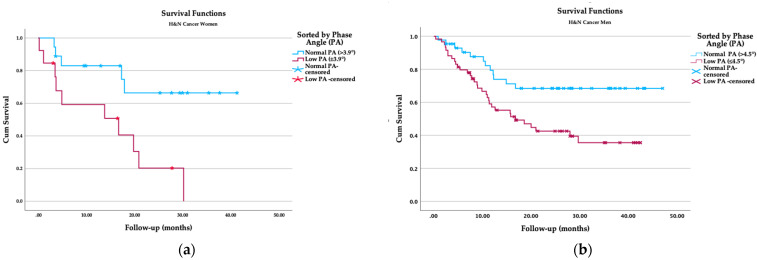
Kaplan–Meier survival analysis: (**a**) survival function by phase angle in women; (**b**) survival function by phase angle in men.

**Table 1 biomedicines-11-01696-t001:** Comparison of clinical characteristics of women and men with H&NC.

	HNC Women (n = 32)	HNC Men (n = 107)	*p* Value
Age *	59 (44–69)	67 (60–73)	0.012
Phase angle *	4.1 (3.3–4.5)	4.5 (3.7–5.1)	0.035
BMI *	26.2 (23.2–27.8)	25.9 (22.1–29.9)	0.706
Handgrip strength *	16.8 (16.2–25.7)	27.2 (24.5–32.9)	<0.001
Four-meter walking speed *	0.79 (0.74–0.97)	0.82 (0.73–0.98)	0.974
SMMI *	6.2 (5.0–7.5)	8.1 (7.03–9.2)	<0.001
Hemoglobin *	12.8 (11.4–13.0)	13.8 (12.4–14.9)	<0.001
Albumin *	4.1 (3.8–4.3)	4.3 (3.9–4.5)	0.241
C-reactive protein *	21.8 (5.5–27.1)	12.3 (3.2–27.1)	0.154
Total body fat % *	41.8 (36.1–47.9)	30.9 (22.3–36.1)	<0.001
Total lymphocyte count *	1618 (1057–1904)	1543 (1191–2150)	0.520
Total cholesterol *	190.0 (151.2–203.5)	184 (152–197)	0.371
Radiotherapy fractions *	33.0 (2.5–34.5)	33 (17.0–35.0)	0.941
Chemotherapy sessions *	3.5 (0–6.0)	1.0 (0–4.0)	0.071
Survivor			0.416
Alive	9 (28.1%)	43 (40.2%)	
Death	15 (46.9%)	45 (42.1%)	
Lost	8 (25.0%)	19 (17.8%)	
Low phase angle	14 (43.8%)	62 (57.9%)	0.157
	Anatomical location		
			0.004
Oral cavity	16 (50%)	27 (25.2%)	
Larynx	3 (9.4%)	51 (47.7%)	
Pharynx	2 (6.3%)	8 (7.5%)	
Salivary gland	3 (9.4%)	3 (2.8%)	
Nose	3 (9.4%)	12 (11.2%)	
Skin	4 (12.5%)	4 (3.7%)	
Other	1 (3.1%)	1 (0.9%)	
Unknown	0	1 (0.9%)	
	Clinical stage		
			0.342
EC I	3 (9.4%)	19 (17.8%)	
EC II	2 (6.3%)	12 (11.2%)	
EC III	7 (21.9%)	18 (16.8%)	
EC IV	14 (43.8%)	49 (45.8%)	
Not available	6 (18.8%)	9 (8.4%)	
Anorexia	12 (37.5%)	29 (27.1%)	
Sarcopenia phenotype			0.093
No sarcopenia	16 (50%)	33 (30.8%)	
Sarcopenia	11 (34.4%)	41 (38.3%)	
Sarcopenic obesity	5 (15.6%)	33 (30.8%)	

* Results are presented as the median and interquartile interval with a significance level <0.05; Mann–Whitney U test. Results are presented in percentages with significance level <0.05; chi-squared test or Fisher’s exact test.

**Table 2 biomedicines-11-01696-t002:** Characteristics of the cohort of women and men with H&NC.

Clinical Characteristic	Phase Angle < 3.9° n (% Inside Specific Group)	Phase Angle ≥ 3.9° n (% Inside Specific Group)	Total n (% Total Patients)	*p* Value
**Women with H&N Cancer (n = 32)**				
Anatomical Location				
Oral cavity	6 (42.9%)	10 (55.6%)	16 (50%)	
Larynx	2 (14.3%)	1 (5.6%)	3 (9.4%)	
Pharynx	2 (14.3%)	0 (0.0%)	2 6.3%)	
Salivary glands	0 (0.0%)	3 (16.7%)	3 (9.4%)	
Nose	2 (14.3%)	1 (5.6%)	3 (9.4%)	
Skin	2 (14.3%)	2 (11.1%)	4 (12.5%)	
Other	0 0(.0%)	1 (5.6%)	1 (3.1%)	
Total	14 (100%)	18 (100%)	32 (100%)	0.296
Histology				
Squamous or epidermoid	10 (71.4%)	12 (66.7%)	22 (68.8%)	
Other	4 (28.6%)	6 (33.3%)	10 (31.3%)	
Total	14 (100%)	18 (100%)	32 (100%)	0.426
Clinical Stage				
I	1 (7.1%)	2 (11.1%)	3 (9.4%)	
II	1(7.1%)	1 (5.6%)	2 (6.3%)	
III	2 (14.3%)	5 (27.8%)	7 (21.9%)	
IV	8 (57.1%)	6 (33.3%)	14 (43.8%)	
Non-classified	2 (14.3%)	4 (22.2%)	6 (18.8%)	
Total	14 (100%)	18 (100%)	32 (100%)	0.717
Phenotype by Body Composition				
No sarcopenia	4 (28.6%)	12 (66.7%)	16 (50.0%)	
Sarcopenia	8 (57.1%)	3 (16.7%)	11 (34.4%)	
Sarcopenic obesity	2 (14.3%)	3 (16.7%)	5 (15.6%)	
Total	14 (100%)	18 (100%)	32 (100%)	0.048
Treatment				
Surgery				
Surgery	4 (28.6%)	13 (72.2%)	17 (53.1%)	
Without surgery	10 (71.4%)	5 (27.8%)	15 (46.9%)	0.017
Radiotherapy				
Radiotherapy	10 (71.4%))	14 (77.7%)	24 (75.0%)	
Without radiotherapy	4 (28.6%)	4 (22.3%)	8 (25.0%)	0.496
Chemotherapy				
Chemotherapy	10 (71.4%)	10 (55.6%)	20 (62.5%)	
Without chemotherapy	4 (28.6%)	8 (44.4%)	12 (37.5%)	0.292
Specific Treatment Modality Combination				
Surgery	1 (7.1%)	1 (5.6%)	2 (6.3%)	
Radiotherapy	0	1 (5.6%)	1 (3.1%)	
Chemotherapy	1 (7.1%)	0 (0%)	1 (3.1%)	
Surgery–radiotherapy	1 (7.1%)	4 (22.2%)	5 (15.6%)	
Surgery–chemotherapy	0	1 (5.6%)	1 (3.1%)	
Chemotherapy–radiotherapy	7 (50%)	2 (11.1%)	9 (28.1%)	
Surgery–chemotherapy–radiotherapy	2 (14.3%)	7 (38.9%)	9 (28.1%)	
Without treatment	2 (14.3%)	2 (11.1%)	4 (12.5%)	
	14 (100%)	28 (100%)	32 (100%)	0.188
**Men with H&NC (n = 107)**				
**Clinical Characteristic**	**Phase angle < 4.5** **°** **n (% inside specific group)**	**Phase angle ≥ 4.5°** **n (% inside specific group)**	**Total n (% total patients)**	***p* Value**
		Anatomical Location		
Oral cavity	22 (35.5%)	5 (11.1%)	27 (25.2%)	
Larynx	26 (41.9%)	25 (55.6%)	51 (47.7%)	
Pharynx	2 (3.2%)	6 (13.3%)	8 (7.5%)	
Salivary glands	1 (1.6%)	2 (4.4%)	3 (2.8%)	
Nose	6 (9.7%)	6 (13.3%)	12 (11.2%)	
Skin	3 (4.9%)	1 (2.2%)	4 (3.7%)	
Other	2 (3.2%)	0 (0.0%)	2 (1.9%)	
Total	62 (100%)	45 (100%)	107 (100%)	0.057
		Histology		
Squamous or epidermoid	59 (95.2%)	42 (93.3%)	101 (94.4%)	
Other	3 (4.8%)	3 (6.7%)	6 (5.6%)	
Total	62 (100%)	45 (100%)	107 (100%)	0.529
		Clinical Stage		
I	8 (12.9%)	11 (24.4%)	19 (17.8%)	
II	8(12.9%)	4 (8.9%)	12 (11.2%)	
III	9 (14.5%)	9 (20.0%)	18 (16.8%)	
IV	33 (53.2%)	16 (35.6%)	49 (45.8%)	
Non-classified	4 (6.5%)	5 (11.1%)	9 (8.4%)	
Total	62 (100%)	45 (100%)	107 (100%)	0.263
Phenotype by Body Composition				
No sarcopenia	13 (21.1%)	20 (44.4%)	33 (30.8%)	
Sarcopenia	29 (46.8%)	12 (26.7%)	41 (38.3%)	
Sarcopenic obesity	20 (32.3%)	13 (28.9%)	33 (30.8%)	
Total	62 (100%)	45 (100%)	107 (100%)	0.023
Treatment				
Surgery				
Surgery	19 (30.6%)	22 (48.9%)	41 (38.3%)	
Without Surgery	43 (69.4%)	23 (51.1%)	66 (61.7%)	0.043
		Radiotherapy		
Radiotherapy	52 (83.9%)	34 (75.6%)	86 (80.4%)	
Without radiotherapy	10 (16.1%)	11 (24.4%)	21 (19.6%)	0.205
		Chemotherapy		
Chemotherapy	37 (59.7%)	21 (46.7%)	58 (54.2%)	
Without chemotherapy	25 (40.3%)	24 (53.3%)	49 (45.8%)	0.128
	Specific Treatment Modality Combination			
Surgery	2 (3.2%)	5 (11.1%)	7 (6.5%)	
Radiotherapy	14 (22.6%)	7 (15.6%)	21 (19.6%)	
Chemotherapy	4 (6.5%)	3 (6.7%)	7 (6.5%)	
Surgery–radiotherapy	7 (11.3%)	9 (20%)	16 (15%)	
Surgery–chemotherapy	2 (3.2%)	0 (0.0%)	2 (1.9%)	
Chemotherapy–radiotherapy	23 (37.1%)	10 (22.2%)	33 (30.8%)	
Surgery–chemotherapy–radiotherapy	8 (12.9%)	8 (17.8%)	16 (15%)	
Without treatment	2 (3.2%)	3 (6.7%)	5 (4.7%)	
Total	62 (100%)	45 (100%)	107 (100%)	0.263

Results are presented in percentages with a significance level <0.05; chi-squared test or Fisher’s exact test.

**Table 3 biomedicines-11-01696-t003:** Anthropometric and biochemical parameters in the cohort of head and neck cancer women and men.

Anthropometrical and Biochemical Indicators	Phase Angle < 3.9° n = 14 Mean (SD Standard Deviation)	Phase Angle ≥ 3.9° n = 18 Mean (SD Standard Deviation)	*p* Value
**Women with H&NC (n = 32)**			
Age and Anthropometrical Indicators			
Age	64.9 (±16.5)	51.8 (±14.8)	0.013
Handgrip strength	17 (±4.5)	22.6 (±7.7)	0.024
Four-meter walking speed	0.7 (±0.3)	0.9 (±0.2)	0.004
Phase angle	3.2 (±0.6)	4.6 (±0.5)	<0.001
Body mass index (BMI)	24.1 (±4.6)	28 (±6.7)	0.071
Total fat percentage	40 (±10)	42 (±7)	0.469
Skeletal muscle mass index (SMMI)	5.5 (±2.5)	7.1 (±1.7)	0.041
Biochemical Indicators			
Hemoglobin	11.8 (±1.9)	12.8 (±0.8)	0.085
Absolute lymphocyte count	1560 (±611) 1507 (1153–17669)	1614 (±753) 1627 (979–1995)	0.830 0.866 **
Albumin **	3.8 (±0.6)	4.3 (±0.3)	0.011
C-reactive protein **	53.5 (±74.5) 27.1 (19–48.5)	15.1 (±12) 10.4 (3.6–27)	0.078 0.027 **
Total cholesterol	192.3 (±38)	187.8 (±43)	0.758
CAR (C-reactive protein (mg/dL)/albumin (g/dL) ratio)	0.66 (0.51–1.20)	0.23 (0.08–0.66)	0.020
Anthropometrical and Biochemical Indicators	Phase angle < 4.5° n = 62 mean (SD standard deviation)	Phase angle ≥ 4.5° n = 45 mean (SD standard deviation)	*p* Value
**Men with H&NC (n = 107)**			
Age and Anthropometrical Indicators			
Age	69 (±11)	60.4 (±10)	<0.001
Handgrip strength	25.3 (±8)	32.9 (±7.2)	<0.001
Four-meter walking speed	0.75 (±0.2)	0.95 (±0.2)	<0.001
Phase angle	3.7 (±0.6)	5.2 (±0.5)	<0.001
Body mass index (BMI)	24.9 (±5.6)	29.1 (±10.4)	0.099
Total fat percentage	29.7 (±10)	29.1 (±7)	0.694
Skeletal muscle mass index (SMMI)	8.6 (±4.6)	10.3 (±5)	0.077
Biochemical Indicators			
Hemoglobin	12.9 (±2)	14.3 (±1.7)	<0.001
Absolute lymphocyte count **	1619 (±991) 1406 (1120–2015)	2023 (±1122) 1739 (1311–2515)	0.052 0.024
Albumin *	4.1 (±0.5)	4.3 (±0.3)	0.001
C-reactive protein **	28.5 (±36.5) 16.3 (5–27)	14.5 (±16.13) 12 (3–21)	0.009 0.022
Total cholesterol	177.4 (±50.2)	188.6 (±30.4)	0.185
CAR (C-reactive protein (mg/dL)/albumin (g/dL) ratio)	0.36 (0.10–0.79)	0.28 (0.07–0.47)	0.015

* Values with parametric distribution at a significance level < 0.05; Student’s *t* test. ** Values with non-parametric distribution expressed in median and interquartile intervals at a significance level < 0.05; Mann–Whitney U test.

**Table 4 biomedicines-11-01696-t004:** Comparison of EORTC QLQ-C30 and EORTC QLQ H&N35 scores between female and male patients with head and neck cancer with low PA and normal PA.

**Scores for the QLQ Scales**	**Phase Angle < 3.9°** **n =14** **Median (P25–P75)**	**Phase Angle ≥ 3.9°** **n =18** **Median (P25–P75)**	***p* Value**
**Women with H&NC**			
**EORTC QLQ-C30 (SCORE 0–100)**			
Global health status/quality of life	58.3 (37.5–77.1)	75 (62.5–85.4)	0.049
Physical functioning	50 (25–81.7)	93.3 (80–100)	<0.001
Role functioning	58.3 (29.2–100)	100 (66.7–100)	0.065
Emotional functioning	66.7 (45.8–85.4)	75 (54.2–91.7)	0.536
Cognitive functioning	66.7 (45.8–100)	100 (83.3–100)	0.091
Social functioning	58.3 (33.3–100)	83.3 (58.3–100)	0.267
Fatigue	61.1 (33.3–61.1)	22.2 (11.1–55.6)	0.014
Nausea and vomiting	0.0 (0.0–16.67)	0.0 (0.0–16.67)	0.896
Pain	16.7 (12.5–83.3)	16.7 (0.0–33.3)	0.235
Dyspnea	16.7 (0.0–41.7)	0.0 (0.0–8.3)	0.180
Insomnia	33.3 (25–75)	16.7 (0.0–41.7)	0.125
Loss of appetite	50 (25–100)	0.0 (0.0–41.7)	0.014
Constipation	33.3 (0.0–66.7)	33.3 (0.0–41.7)	0.536
Diarrhea	0.0 (0.0–33.3)	0.0 (0.0–0.0)	0.464
Financial difficulties	33.3 (0.0–66.7)	33.3 (0.0–66.7)	0.613
**EORTC QLQ H&N35 (SCORE 0–100)**			
Pain	33.3 (22.9–66.7)	24.75 (8.3–41.7)	0.084
Swallowing	33.3 (14.6–64.6)	4.2 (0.0–25)	0.020
Sensory problems	25 (0.0–71)	0.0 (0.0–20)	0.125
Speech problems	44.4 (19.4–72.2)	22.2 (0.0–42.2)	0.077
Trouble with social eating	25 (0.0–79.2)	16.7 (0.0–29.2)	0.235
Trouble with social contact	33.3 (0.0–68.3)	0.0 (0.0–11)	0.030
Less sexuality	41 (36.6–41)	41 (41–41)	0.419
Teeth	100 (0.0–100)	10.3 (0.0–33.3)	0.020
Opening mouth	83. (0.0–100)	28 (0.0–33.3)	0.116
Dry mouth	33.3 (0–100)	33.3 (0.0–66.7)	0.536
Sticky saliva	100 (0.0–100)	0.0 (0.0–35.3)	0.027
Coughing	33.3 (0.0–66.7)	0.0 (0.0–66.7)	0.587
Felt ill	33.3 (0.0–100)	0.0 (0.0–33.3)	0.180
Pain killers	100 (100–100)	100 (0–100)	0.168
Nutritional supplements	0.0 (0.0–100)	0.0 (0.0–25)	0.145
Feeding tube	0.0 (0.0–100)	0.0 (0.0–100)	0.419
Weight loss	100 (0.0–100)	100 (0.0–100)	0.750
Weight gain	0.0 (0.0–0.0)	0.0 (0.0–100)	0.267
**Scores for the QLQ Scales**	**Phase Angle < 4.53°** **n = 62** **Median (P25–P75)**	**Phase Angle ≥ 4.53°** **n = 45** **Median (P25–P75)**	***p* Value**
**Men with H&NC**			
**EORTC QLQ-C30 (SCORE 0–100)**			
Global health status/quality of life	69.8 (50–83.3)	83.3 (66.7–100)	0.006
Physical functioning	73.3 (45–88.3)	93.3 (80–100)	<0.001
Role functioning	83.3 (33.3–100)	100 (75–100)	0.032
Emotional functioning	66.7 (50–91.7)	83.3 (66.7–91.7)	0.293
Cognitive functioning	83.3 (66.7–100)	100 (66.7–100)	0.137
Social functioning	83.3 (50–100)	100 (66.7–100)	0.130
Fatigue	44.4 (19.4–66.7)	11.1 (0.0–33.3)	<0.001
Nausea and vomiting	0.0 (0.0–0.0)	0.0 (0.0–8.3)	0.667
Pain	16.7 (0.0–50)	0.0 (0.0–33.3)	0.084
Dyspnea	0.0 (0.0–33.3)	0.0 (0.0–33.3)	0.146
Insomnia	33.3 (0.0–100)	33.3 (0.0–50)	0.071
Loss of appetite	0.0 (0.0–33.3)	0.0 (0.0–0.0)	0.013
Constipation	16.6 (0.0–33.3)	0.0 (0.0–33.3)	0.748
Diarrhea	0.0 (0.0–0.0)	0.0 (0.0–0.0)	0.483
Financial difficulties	33.3 (0.0–66.7)	33.3 (0.0–66.7)	0.375
**EORTC QLQ H&N35 (SCORE 0–100)**			
Pain	25 (6.2–58.3)	24.5 (0.0–29.2)	0.086
Swallowing	25 (6.2–50)	8.3 (0.0–29)	0.010
Sensory problems	16.7 (0.0–50)	0.0 (0.0–30)	0.023
Speech problems	37.8 (11–66.7)	33.3 (11–55.6)	0.640
Trouble with social eating	19.4 (0.0–41.7)	8.3 (0.0–25)	0.031
Trouble with social contact	6.7 (0.0 21.7)	6.7 (0.0 11.8)	0.613
Less sexuality	41 (41–41)	41 (35.6–41)	0.051
Teeth	0.0 (0.0–33.3)	0.0 (0.0–33.3)	0.550
Opening mouth	23 (0.0–66.7)	0.0 (0.0–28)	0.043
Dry mouth	33.3 (0.0–66.7)	33.3 (0.0–54)	0.194
Sticky saliva	33.3 (0.0–66.7)	33.3 (0.0–54)	0.168
Coughing	33.3 (0.0–66.7)	27.4 (0.0–33.3)	0.475
Felt ill	25 (0.0–66.7)	0.0 (0.0–33.3)	0.274
Pain killers	100 (0.0–100)	0.0 (0.0–100)	0.022
Nutritional supplements	10.5 (0.0–100)	0.0 (0.0–60.5)	0.117
Feeding tube	0.0 (0.0–0.0)	0.0 (0.0–0.0)	0.350
Weight loss	100 (0–100)	34.3 (0–100)	0.515
Weight gain	0.0 (0.0–0.0)	0.0 (0.0–4.5)	0.074

Significant *p* value < 0.05; Mann–Whitney U test.

**Table 5 biomedicines-11-01696-t005:** Survival, death, and loss to follow-up.

Survival Status				
Women with H&NC (n = 32)				
Clinical Characteristic	Phase Angle < 3.9° n = 14 (% in a Specific Group)	Phase Angle ≥ 3.9° n = 18 (% in a Specific Group)	Total n = 32 (% Total Patients)	*p* Value
Alive	2 (14.3%)	8 (44.4%)	10 (31.3%)	
Death	10 (71.4%)	5 (27.8%)	15 (46.9%)	
Lost to follow-up	2 (14.3%)	5 (27.8%)	7 (21.8%)	
Total	14 (100%)	18 (100%)	32 (100%)	0.046 *
Survival Time for Women				
Mean (95% CI mean) survival time in women (months)	14.25 (7.8–20.7)	30.9 (23.5–38.5)	23.7 (17.6–29.8)	**0.007 ****
**Survival Status**				
**Men with H&NC (n = 107)**				
**Clinical Characteristic**	**Phase Angle < 4.5** **°** **n = 62 (% in a Specific Group)**	**Phase Angle ≥ 4.5°** **n = 45 (% in a Specific Group)**	**Total n = 107 (% Total Patients)**	** *p* ** **V** **alue**
Alive	23 (37.1%)	25 (55.5%)	48(44.9%)	
Death	33 (53.2%)	12 (26.7%)	45 (42.1%)	
Lost to follow-up	6 (9.7%)	8(17.8%)	14 (13.1%)	
Total	62 (100%)	45 (100%)	107 (100%)	0.022 *
Survival Time for Men				
Mean (95% CI mean) survival time in men (months)	22.5 (18–27)	35.2 (29.5–40.8)	28.7 (24.7–32.6)	0.008 ** 0.001 ***

* Significant *p* value < 0.05; chi-squared test. ** Significant *p* value < 0.05; log rank (Mantel–Cox); *** *p* value for differences in survival log rank between men and women.

## Data Availability

The datasets generated and/or analyzed during the current study are not publicly available because they are the property of the Instituto Mexicano del Seguro Social. Institutional and federal dispositions restrict unlimited access to personal data, but they are available from the corresponding authors upon reasonable request with prior authorization from the institution.

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
