# Peer review of "Phase Angle in Head and Neck Cancer: A Sex-Differential Analysis from Biological and Clinical Behavior to Health-Related Quality of Life"

_biomedicines, 2023, doi:10.3390/biomedicines11061696_

Round 1

Reviewer 1 Report

General comments:

The paper is of potential scientific interest; however, a thorough revision is required before being considered for publication.

The Introduction requires a better, more logical flow that leads the reader to the main goal of this work: HNC incidence and causes/risk factors – treatment types – normal tissue effects – gender differences – aim of the work.

Introduction – line 137 – ‘H&NC treatments impair nutritional status…’. The authors must include a paragraph on the treatment of HNC that impacts their QoL. Radiotherapy, surgery, chemotherapy and immunotherapy all have different effects which should be mentioned here.

Introduction – since the goal of the article is to analyse a sex-dependent treatment outcome among HNC patients, there must be mention of the current knowledge on sex-specific response to radiotherapy (see for instance: https://pubmed.ncbi.nlm.nih.gov/31991224/ ; https://www.redjournal.org/article/S0360-3016(17)34461-9/fulltext) or chemotherapy (https://pubmed.ncbi.nlm.nih.gov/25214172/;  https://pubmed.ncbi.nlm.nih.gov/35215367/)

Define phase angle in the introduction.

Methods: include a table with patient characteristics (mean age (range), tumour location, sex, stage, etc).

Divide the Methods section into (1) Patient characteristics, (2) Measurements for anthropometric analysis, (3) Statistical analysis.

Results: the paragraphs from line 242 to 248 should be included in the Methods – patient characteristics.

Table 1 should be divided into Table x. Clinical characteristics of patients (and included in Methods as specified above – age, sex, treatment, anatomical location, staging) and Table y. Anthropometric parameters (the rest of parameters) – included in the Results as these were measurements of your study.

Often the writing style is tortuous and/or grammatically incorrect. Therefore, the English language must be revised throughout the manuscript, preferably by a native speaker.

Specific comments:

1.     Line 102 – ‘risk factors such as …’

2.     Line 104 – the following statement requires referencing: ‘significant range of young white males having a higher risk of developing H&NC than other groups’

3.     Line 107 – rephrase - ‘The Globocan 2020 report on the global burden…’

4.     Line 112 – ‘…to understand the role of sex-specific biological characteristics in disease development…’

5.     Lines 114-117 – this paragraph is repetitive; it was mentioned previously in the Introduction.

6.     Line 140 – ‘The high metabolic rate of this cancer results in malnutrition-related weight loss..’

7.     Line 141 – the treatment is obviously not ‘optimal’ given the side effects. Replace ‘optimal’ with ‘the best available’ or along these lines.

8.     Line 164 – ‘…are the reasons behind determining the sex-specific variation of the PA as a predictor…’

9.     Lines 297, 303, table 4 (two instances) – replace ‘physic functioning’ with ‘physical functioning’

10.  Line 323/324 – the sentence is linguistically incorrect, please rephrase.

11.  Line 326 – replace ‘reflexed’ with ‘reflected’.

12.  Line 343 – replace ‘have been affected’ with ‘are affected’.

13.  Line 346 – ‘modification’ of what?

14.  Line 347 – ‘that condition metabolic…’

15.  Line 425 – ‘…also impacting swallowing, … all of these factors being related to functionality in women’.

16.  Line 429 – ‘…and, similarly to the female group, fatigue…’

17.  Line 431 – ‘…other aspects involved are sensory issues, trouble with social eating, opening their mouth, and pain management, reaching higher scores than women’.

The paper requires a comprehensive English language revision.

Author Response

Comments and Suggestions for Authors

General comments:

The paper is of potential scientific interest; however, a thorough revision is required before being considered for publication.

The Introduction requires a better, more logical flow that leads the reader to the main goal of this work: HNC incidence and causes/risk factors – treatment types – normal tissue effects – gender differences – aim of the work.

Introduction – line 137 – ‘H&NC treatments impair nutritional status…’. The authors must include a paragraph on the treatment of HNC that impacts their QoL. Radiotherapy, surgery, chemotherapy and immunotherapy all have different effects which should be mentioned here.

Answer: We re-wrote the introduction section, and we added some paragraphs describing H&NC incidences, causes, risk factors, treatment types, the effect of treatment on normal tissues and sex differences. We emphasized about the aim of the manuscript; the changes are highlighted in green colour.  

Introduction – since the goal of the article is to analyse a sex-dependent treatment outcome among HNC patients, there must be mention of the current knowledge on sex-specific response to radiotherapy (see for instance: https://pubmed.ncbi.nlm.nih.gov/31991224/ ; https://www.redjournal.org/article/S0360-3016(17)34461-9/fulltext) or chemotherapy (https://pubmed.ncbi.nlm.nih.gov/25214172/;  https://pubmed.ncbi.nlm.nih.gov/35215367/)

Answer: We reviewed the suggested articles, and we took some of them to reference in the text, adding other complementary bibliography.

Define phase angle in the introduction.

Answer: We described in a more technical way the phase angle concept.

Methods: include a table with patient characteristics (mean age (range), tumour location, sex, stage, etc).

Divide the Methods section into (1) Patient characteristics, (2) Measurements for anthropometric analysis, (3) Statistical analysis.

Answer: We decided to follow the template instructions provided by the journal, and as part of those instructions we might not include tables in methods section. However, we decided to include headings and subheadings with:

  • Patients’ selection criteria
  • Anthropometric analysis
  • Statistical analysis

Results: the paragraphs from line 242 to 248 should be included in the Methods – patient characteristics.

Answer: The paragraphs and table remain in the results section as it is described in the instructions contained in the journal template

Table 1 should be divided into Table x. Clinical characteristics of patients (and included in Methods as specified above – age, sex, treatment, anatomical location, staging) and Table y. Anthropometric parameters (the rest of parameters) – included in the Results as these were measurements of your study.

Answer: We did not separate the table for the reasons above-mentioned

Often the writing style is tortuous and/or grammatically incorrect. Therefore, the English language must be revised throughout the manuscript, preferably by a native speaker.

Answer: We are sending the paper to style review through MDPI author services platform.

Specific comments:

  1. Line 102 – ‘risk factors such as …’
  2. Line 104 – the following statement requires referencing: ‘significant range of young white males having a higher risk of developing H&NC than other groups’
  3. Line 107 – rephrase - ‘The Globocan 2020 report on the global burden…’
  4. Line 112 – ‘…to understand the role of sex-specific biological characteristics in disease development…’
  5. Lines 114-117 – this paragraph is repetitive; it was mentioned previously in the Introduction.
  6. Line 140 – ‘The high metabolic rate of this cancer results in malnutrition-related weight loss..’
  7. Line 141 – the treatment is obviously not ‘optimal’ given the side effects. Replace ‘optimal’ with ‘the best available’ or along these lines.
  8. Line 164 – ‘…are the reasons behind determining the sex-specific variation of the PA as a predictor…’
  9. Lines 297, 303, table 4 (two instances) – replace ‘physic functioning’ with ‘physical functioning’
  10. Line 323/324 – the sentence is linguistically incorrect, please rephrase.
  11. Line 326 – replace ‘reflexed’ with ‘reflected’.
  12. Line 343 – replace ‘have been affected’ with ‘are affected’.
  13. Line 346 – ‘modification’ of what?
  14. Line 347 – ‘that condition metabolic…’
  15. Line 425 – ‘…also impacting swallowing, … all of these factors being related to functionality in women’.
  16. Line 429 – ‘…and, similarly to the female group, fatigue…’
  17. Line 431 – ‘…other aspects involved are sensory issues, trouble with social eating, opening their mouth, and pain management, reaching higher scores than women’.

Answer: All the specific modifications were made, excepting for the paragraphs that were deleted.

Comments on the Quality of English Language

The paper requires a comprehensive English language revision.

Answer: We are sending the paper to style review through MDPI author services platform.

Reviewer 2 Report

Dear collegues.

Dear colleagues!

I do not entirely agree with parts of your introduction:

in line 107-112 you point out the incidences (men 2/3, women 1/3). In line 122 you say that women are underrepresented and consequently treatment in the early stages is missing.

This conclusion does not make sense.

Line 135: Here, stage-dependent survival rates (OS) of other tumour entities are missing in comparison to head and neck tumours, so that this (your) statement is valid.

Line 140: attributing weight loss to a high metabolic rate alone is misleading. Most patients have effective swallowing difficulties as a result of the tumour in the upper swallowing tract. This fact should not be left as it is.

From line 313:

Survival status: You write that the women with PA < 3.9° have a shorter survival overall, yet more patients with stage III/IV tumours are found in this group (70% , compared to 60% in PA >3.9; see Table 2.) A similar constellation is found in the group of male tumour patients. In my view, there are other parameters that explain the shorter survival time. This must be explained! There is a selection BIAS here.

Chapter 3.4.2.: here it should read: men (the results of women are discussed in chapter 3.4.1.)

Line 363: the conclusion that women perform worse than men depends, in my opinion, more on the tumour stage and tumour location than is described in your assessment. In women, tumours of the oral cavity predominate, which per se already lead to mechanical dysphagia; in men, laryngeal carcinomas predominate (51 cases), which again leads to a selection BIAS.

In summary, it is certainly scientifically interesting to collect data on nutrition (BMI, PA, SMMI, etc.) and correlate them with QoL, OS and functional outcome in tumour patients.

Nevertheless, the selected patient group seems to me to be too inhomogeneous, e.g. with regard to age, tumour location and tumour stage when comparing women and men, to be able to derive clear conclusions. 

In my opinion, there is a selection bias here that should not be neglected.

Best regards

Author Response

Comments and Suggestions for Authors

I do not entirely agree with parts of your introduction:

in line 107-112 you point out the incidences (men 2/3, women 1/3). In line 122 you say that women are underrepresented and consequently treatment in the early stages is missing.

Answer: We changed the introduction section, and we re-wrote it, considering your comments and the other reviewer comments.

This conclusion does not make sense.

Line 135: Here, stage-dependent survival rates (OS) of other tumour entities are missing in comparison to head and neck tumours, so that this (your) statement is valid.

Answer: According to the distribution per clinical stage the major proportion of patients were in CS III and IV, in both male and female (62.6% and 65.7% respectively) and during multivariate analysis the CS appeared as a confounding variable.

Line 140: attributing weight loss to a high metabolic rate alone is misleading. Most patients have effective swallowing difficulties as a result of the tumour in the upper swallowing tract. This fact should not be left as it is.

Answer: We re-wrote the paragraph, because we recognize that you are right about the specific issue you signalized, it is a multifactorial condition where the metabolic rate and swallowing difficulties, joined to a pro-inflammatory and tumor cachexia state, interfere with the body weight maintenance.

From line 313:

Survival status: You write that the women with PA < 3.9° have a shorter survival overall, yet more patients with stage III/IV tumours are found in this group (70% , compared to 60% in PA >3.9; see Table 2.) A similar constellation is found in the group of male tumour patients. In my view, there are other parameters that explain the shorter survival time. This must be explained! There is a selection BIAS here.

Answer: During the first steps of the multivariate analysis, we did not find significant correlation between clinical stage and mortality rate, and the observed differences in the percentage of patients in clinical stage III and IV were not statistically significant. The sample was obtained through consecutive sampling technique, and we included all incident patient that attended to the facility center. We did not exclude any patient.

Chapter 3.4.2.: here it should read: men (the results of women are discussed in chapter 3.4.1.)

Answer: We corrected the typo mistake.

Line 363: the conclusion that women perform worse than men depends, in my opinion, more on the tumour stage and tumour location than is described in your assessment. In women, tumours of the oral cavity predominate, which per se already lead to mechanical dysphagia; in men, laryngeal carcinomas predominate (51 cases), which again leads to a selection BIAS.

Answer: It is true that in women group the oral cavity cancers were predominant, as well as the laryngeal carcinomas were in men. However, most of the patients were treated with radiotherapy and chemoradiotherapy and surgery at different moments of their disease evolution, and all these therapeutic interventions affect the aerodigestive tract, conditioning eating problems related to dysphagia, and swallowing difficulties associated to mucositis, wound healing issues, fistulae, etcetera.

In summary, it is certainly scientifically interesting to collect data on nutrition (BMI, PA, SMMI, etc.) and correlate them with QoL, OS and functional outcome in tumour patients.

Nevertheless, the selected patient group seems to me to be too inhomogeneous, e.g. with regard to age, tumour location and tumour stage when comparing women and men, to be able to derive clear conclusions. 

In my opinion, there is a selection bias here that should not be neglected.

Answer: We included some biological characteristics not related to clinical stage, anatomical location, age or sex, such as C-reactive protein values and the CAR (C-reactive protein/albumin ratio) that indicates tumor cachexia and an important inflammatory state in patients with low phase angle (<3.9 for female and <4.5 for males) in a strong correlation with mortality, the functional outcome and the poor prognosis and worse quality of life.

This manuscript has not been published elsewhere and is not under consideration by another journal. We have approved the manuscript and agree with submission to the special issue “Head and Neck Tumors 2.0" in the Journal “Biomedicines”.

We confirm that neither the manuscript nor any parts of its content are currently under consideration or published in another journal. All authors have approved the manuscript and agree with its submission to the Journal “Biomedicines”, and there are no conflicts of interest to declare.

The manuscript has been carefully reviewed. This study's findings are relevant to your journal's scope and will be of interest to its readership. We look forward to hearing from you at your earliest convenience.

Round 2

Reviewer 1 Report

The authors have addressed most comments raised by this reviewer.

Reviewer 2 Report

I have no further comments